# The Acute Achilles Tendon Rupture: An Evidence-Based Approach from the Diagnosis to the Treatment

**DOI:** 10.3390/medicina58091195

**Published:** 2022-09-01

**Authors:** Francesco Amendola, Léa Barbasse, Riccardo Carbonaro, Mario Alessandri-Bonetti, Giuseppe Cottone, Michele Riccio, Francesco De Francesco, Luca Vaienti, Kevin Serror

**Affiliations:** 1Plastic and Reconstructive Surgery Department, I.R.C.C.S. Istituto Ortopedico Galeazzi, 20122 Milan, Italy; 2Plastic and Reconstructive Department, AP-HP, Hôpital Saint-Louis, 75010 Paris, France; 3Hand Surgery Unit, Department of Plastic and Reconstructive Surgery, Azienda “Ospedali Riuniti”, Via Conca 21, 60126 Ancona, Italy

**Keywords:** achilles tendon rupture, operative treatment, minimally invasive surgery, open repair, nonoperative treatment

## Abstract

*Background and Objective:* Acute Achilles tendon rupture (AATR) is a common injury with a significant impact on daily living. Although various systematic reviews and meta-analyses have been written on the topic, no actual consensus exists on the best treatment. We aimed to collect the highest quality of evidence on the subject and to produce a document to which to refer, from the diagnosis to the final treatment. *Material and Methods:* Inclusion criteria were systematic reviews discussing Achilles tendon rupture, concerning either diagnostic criteria, classification, or treatment; English language; clearly stated inclusion and exclusion criteria for patients’ selection. *Results:* Thirteen systematic reviews were included in the study. A strong consensus exists about the higher risk of re-rupture associated with non-operative treatment and a higher risk of complications associated with surgical repair. *Conclusions:* The combination of minimally invasive repair and accelerated functional rehabilitation seems to offer the best results in the treatment of Achilles tendon rupture.

## 1. Introduction

Acute Achilles’ tendon rupture (AATR) is a common injury, with an annual incidence of 5 to 50 events per 100,000 people and may result in severe disability and prolonged absence from work and physical activity. With more than 10,000 references presently published on the topic, and more than 200 systematic reviews, debate still exists about the best treatment strategy and the rehabilitation protocol. Meta-analyses including different trials at different times, and focusing on different outcomes, further increased the confusion on the topic.

Different treatment strategies have been proposed over the years, mainly categorized in open repair (OR), minimally invasive surgery (MIS) and nonoperative treatment (NOT). OR requires about 10 cm of vertical posteromedial incision and wide tendon exposure. The supposed advantages of OR are the lowest re-rupture rate and fastest return to sports activity while showing the highest rate of complications. MIS includes different techniques (Ma and Griffith’s technique [1], Dresden technique [2], Tenolig [3] and Achillon [4]), all with a reduced incision length, minimizing the exposure of the Achilles tendon and, theoretically, lowering the complications while preserving the same efficacy on the prevention of re-ruptures. Eventually, the NOT has shown the lowest complications rate, but requires longer healing time and thus increased disability.

Several meta-analyses are present in the literature, showing different complications and re-rupture rates among the three different techniques. Considering only systematic reviews of clinical trials, it becomes evident how each of them includes different trials and bases its results on different studies. Our work aimed to systematically collect all the actual Level 1 quality of evidence on this topic and to perform a critical analysis of the various treatment strategies. Moreover, we also analyzed the high-quality evidence regarding risk factors, clinical predictors and rehabilitation protocols. Therefore, our purpose is to provide a complete and updated reference for the diagnosis and treatment of acute Achilles tendon rupture.

## 2. Materials and Methods

We searched Medline and Embase databases for “Achilles tendon”, to collect as many references as possible. The results were filtered by systematic review and meta-analysis, and duplicates were removed. The remaining were screened by abstract. Inclusion criteria were studies on diagnostic or therapeutic options for acute Achilles tendon rupture; systematic reviews clearly defining the research strategy and following the Preferred Reporting Items for Systematic review and Metanalysis (PRISMA) statement. Exclusion criteria were studies focused on Achilles tendinopathy or chronic tendinitis; studies on animals; systematic reviews including case series or case reports. The included articles were divided into three main categories, based on their main subject: risk factors, diagnostics and therapeutics. After reading and analysing every article included, a selected number of clinical trials of outstanding clinical importance were also included in this review and cited throughout the manuscript. The review was registered in the international database of prospectively registered systematic reviews.

## 3. Results

Searching for “Achilles tendon”, a total of 4209 Medline only, 4805 Embase and 8216 common references appeared. After screening for systematic review and meta-analysis, 270 studies in Medline and 407 in Embase were screened by abstract and included according to the above-mentioned criteria. Finally, 13 systematic reviews were included in the present studies. The selection process is outlined in Figure 1. Two studies discussed the risk factors of AATR [5] and the role of ultrasound analysis [6], one study performed a cost-effectiveness analysis [7], seven studies discussed the treatment of AATR [8,9,10,11,12,13,14], and three studies discussed the rehabilitation protocol [15,16,17].

We summarized the total citations and different study types resulting from each keywords search, as shown in Table 1 and Figure 2.

### 3.1. Clinical Risk Factors

Claessen et al. [5] performed a systematic review investigating predictive factors influencing Achilles tendon rupture. They included 31 studies, of which only two were high-quality evidence studies. No randomized clinical trials (RCTs) were present in the literature, and the studies included were all case-control or cross-sectional studies. They distinguished among non-modifiable (age, race, comorbidities, genetic expression) and modifiable risk factors (weight, drug prescriptions and lifestyle). Among the non-modifiable determinants, limited evidence showed an increased risk of tendon rupture for the male sex, higher age, Black race, different genes expressions (i.e., COX2, OSM, LIF, IL6, IL6R, VEGF), modifications in collagen type and content (more type III collagen, less type I collagen and less overall collagen content). Particular comorbidities are also associated with an increased risk of AATR (i.e., renal transplant, inflammatory bowel disease, ipsilateral sciatic pain, trauma, autoimmune arthritis, infectious arthritis, rheumatoid arthritis, and previous tendinopathy). Interestingly, a higher anteroposterior diameter of the Achilles tendon was also associated with an increased risk of rupture. Based on animal studies, the loss of larger fibrils in the core and periphery of the tendon was the single most important non-modifiable risk factor for rupture. On the other hand, limited evidence was found for BMI > 25, oral quinolone use, oral corticosteroid use, living in urban areas and hyper-cholesterolemia. Conflicting evidence and low-quality studies suggested a correlation with physical activity, both professional and recreational. In conclusion, the decreased tendon fibril size can be considered the only proven predictor for tendon rupture.

### 3.2. Diagnosis

Physical examination for the diagnosis of complete Achilles tendon rupture using several diagnostic maneuvers—including the Thompson test, decreased resting tension of the tendon while prone, and the presence of a palpable tendon defect—showed a sensitivity between 73% and 96%. Aminlari et al. [6] performed a systematic review on the efficacy of ultrasound analysis in the early diagnosis of Achilles tendon rupture. They reported on a total of 15 studies with 808 patients, including only original studies with at least five patients with a sonographic diagnosis of Achilles tendon rupture (complete or partial). They compared the diagnostic accuracy of ultrasound analysis and surgery, as the reference standard. For complete tendon rupture, the ultrasound showed an overall sensitivity and specificity of 95% and 99%, respectively. Comparable results were demonstrated for partial ruptures (94% and 97%). Statistical analysis confirmed a low risk of bias and high concordance among studies included in the meta-analysis. Data from this meta-analysis suggest that a negative Achilles tendon ultrasound (i.e., the absence of rupture) rules out with high confidence a complete Achilles tendon rupture. The high sensitivity with narrow confidence intervals (CIs) suggested that a normal Achilles tendon on ultrasound (i.e., a negative ultrasound) implies a very low likelihood that the patient has a complete or partial Achilles tendon rupture.

### 3.3. Operative versus Nonoperative Treatment

We included seven systematic reviews comparing operative versus nonoperative treatment of the acute Achilles tendon rupture.

In a 2010 Cochrane systematic review of clinical trials done by Khan and Smith [13], surgical treatment was compared to NOT in adult patients with AATR. They included six trials, for a total of 566 patients, confirming that the surgical treatment has lower re-rupture rates (RR 0.41) but higher complication rates (RR 4.89). Similarly, they also compared MIS to OR, with the latter resulting in higher complication rates compared to MIS (RR 9.32). However, reported results are tempered by the low number of patients enrolled in the trials (n. 174).

Ochen et al. [14] performed a systematic review and meta-analysis of 10 trials (including 944 patients), and 19 observational studies (including 14,918 patients). The authors decided to include also observational studies in their meta-analysis to increase sample size, which could enable the evaluation of small treatment effects and infrequent outcome measures. Furthermore, they considered that observational studies might provide insight into a variety of populations and long-term effects compared with the usually highly selected patient populations in RCTs. Operative treatment showed a significantly reduced risk of re-rupture compared to non-operative treatment (2.3% vs. 3.9%), both in overall pooled analysis (Risk ratio 0.43) and RCT/Observational pooled analyses (0.40 and 0.42, respectively). On the other hand, their analysis showed an increased risk of complications (wound infection, deep vein thrombosis DVT, sural nerve injury) in the operative group compared to the non-operative group (4.9% vs. 1.6%). It must be noted that most complications in OR were wound infections (2.8%), while the main complication in NOT was DVT (1.2%). No differences were found in functional outcome and return to sport or work between the two approaches.

Wu et al. [11] compared different treatment strategies (OR, MIS and NOT) combined with different rehabilitation protocols (accelerated rehabilitation AR and early immobilization EI). Twenty-nine RCTs were included, for a total of 2060 patients. They reported a major complications rate of 9.13%, with a 5% re-rupture rate and a 1.5% incidence of deep infections. They concluded that the treatment association of NOT + EI had a significantly higher major complications rate compared to others. The combination of MIS + AR resulted to be the safest strategy of treatment, with the lowest risk of re-ruptures, deep vein thromboses and infections.

Shi et al. [8] performed a network meta-analysis including 38 randomized controlled trials involving 2480 participants and comparing 6 therapeutic regimens: open repair (OR), minimally invasive surgery (MIS) and nonoperative treatment (NOT) combined with traditional standard rehabilitation (TSR) and accelerated functional rehabilitation (AFR). The main outcomes considered were re-rupture rate, wound-related complication (wound/skin infection, scar/skin adhesion), sural nerve injury, deep venous thrombosis (DVT) and the number of patients returning to sport. They conclude that the operative treatment (without distinction between open repair and minimally invasive repair) has lower re-rupture rates compared to nonoperative treatment; these advantages are enhanced when an AFR protocol is associated with the surgery. On the other hand, MIS and AFR are significantly associated with lower wound complication rates and faster return to sport compared to both OR/NOT and TSR, respectively. Deep vein thrombosis and sural nerve injury rates were not significantly different among the techniques.

Su et al. [7] performed a cost-effectiveness analysis comparing operative treatment against non-operative treatment. They ran a sensitivity analysis based on actual direct cost data, expected utility values from a previously published decision analysis, and outcome probabilities from published systematic reviews. They reported an average cost for OR treatment of USD 12,477 versus USD 3100 for NOT treatment. Notwithstanding NOT was more cost-effective, the operative treatment became increasingly cost-effective as the utility of well-being increased: for example, with high-demand patients, such as athletes, or those in physically demanding occupations who prefer the highest possible functional outcome as early as possible. Similarly, Westin et al. performed a cost-effectiveness analysis considering quality-adjusted life years (QALY), as a measurement that combines the health-related quality of life and life expectancy in one metric. They concluded that surgical treatment is more expensive but is also associated with a slightly better health outcome, with a resulting cost per QALY gained of EUR 45,855.

Attia et al. [9] performed a meta-analysis of RCTs comparing OR versus MIS, in terms of functional outcomes and complications, again including the American Orthopaedic Foot & Ankle Society (AOFAS) score, Achilles tendon Total Rupture Score (ATRS) score, sural nerve injuries, infections, skin complications, re-ruptures, ankle range of motion and calf circumference. No significant differences were found in the AOFAS score and rate of return to sports activity between the two techniques. MIS registered a small but significantly higher ATRS compared to OR. The overall complications rate did not differ among the two techniques, but the MIS group was associated with a significantly higher sural nerve injury rate and the OR was associated with a higher superficial infection rate. Eventually, MIS was associated with wider plantar flexion and lower ankle stiffness, compared to OR. They included and expanded the meta-analysis performed by Grassi et Al. [10], who already showed no differences in terms of re-ruptures between MIS and OR (RR 0.64), but the MIS was associated with lower complications (RR 0.18) and lower infections (RR 0.15).

Seow and colleagues [12] performed a similar meta-analysis on different treatments of AATR, including surgical, non-surgical and mini-invasive approaches. They compared the techniques, including also all the patients that were lost at the follow-up (and subsequently were excluded from the analyses in the involved studies). Then, they considered whether these patients had experienced a re-rupture (worst case scenario) or had not (best case scenario). They showed how many clinical studies on this topic, including trials and meta-analyses, were, in fact, under-powered. Being the re-rupture a dichotomous variable, a large population in individual clinical studies is required to have sufficient statistical power to detect a difference between the treatment arms. This enhanced the advantage of performing a meta-analysis on the topic because only with a pooled analysis, the study might reach sufficient power. In their pooled analysis, the surgical treatment showed to be superior to the NOT in terms of re-rupture, while the latter was superior in terms of infections and complications. They also confirmed that the MIS resulted in similar rates of re-ruptures, but significantly fewer complications, compared to OR. Surprisingly, there was no difference in the results whether early or later rehabilitation was chosen after either nonsurgical or surgical treatment.

### 3.4. Rehabilitation Therapy

Zellers and colleagues [17] performed a meta-analysis to define the Early Rehabilitation therapy protocol, which, in literature, is vague. They concluded that the ERT is initiated within two weeks from the surgical intervention or cast application, and mainly includes weight-bearing initiated within the first week, and exercises (e.g., ankle range of motion, strengthening, whole-body conditioning) initiated in the second week. Ghaddaf et al. [16] similarly performed a meta-analysis comparing early (<4 weeks) vs. late weight-bearing in patients with AATR, concluding that there was no significant difference between early WB and late WB in terms of re-rupture rate, return to pre-injury sport activity, time to return to work or adverse event rate. The same results were confirmed by Zhang et al. [15].

## 4. Discussion

Controversies still exist about the clinical risk factors and treatment strategies of the AATR. With thousands of references present in the literature, and many meta-analyses already performed about different aspects of this topic performed in different times, these controversies emerge now more than ever. A single summary of the actual evidence, from the risk factors to the best treatment strategy, with critical analysis, is the aim of the present study.

About the clinical risk factors, male sex, higher age and Black race showed a significant, but limited effect, on the risk of AATR. On the contrary, and very remarkably, limited evidence was found for what was thought to be the most common risk factors, such as BMI > 25, oral quinolone use, oral corticosteroid use, living in urban areas and hyper-cholesterolemia. Conflicting evidence and low-quality studies were present about physical activity, both professional and recreational. In the end, the decreased tendon fibril size can be considered the only true clinical predictor of the risk of tendon rupture.

Only thirteen systematic reviews were included in this study, among the over 200 appearing after searching for “Achilles tendon” in different databases. The seven systematic reviews about the treatment strategies cited a total of 51 different randomized clinical trials on the subject, and the most interesting part is that there was little accordance on the included RCTs among the different SRs, as shown in (Table 2).

None of the RCTs was included in all systematic reviews. Only one RCT [35] was present in six out of the seven SRs and only seven RCTs [24,30,32,33,40,41,64] were included in five SRs. Basically, the actual clinical indications for the AATR treatment are based on these eight RCTs.

All the SRs included in our study report a higher rate of re-ruptures (3–13% vs. 1–5%) for the non-operative treatment and a higher rate of overall complications other than re-rupture (0–13% vs. 4–18%) and infections (0–1% vs. 1–6%) for the operative treatment. Similarly, the minimally invasive surgery always results associated with lower complications rates (7–10% vs. 5–18%) compared with the open repair, while showing a comparable rate of re-ruptures (1–5% vs. 2–5% and infections (0–6% vs. 1–6%). In active young men, the early functional recovery outweighs the risk of complications such as infections, for which different treatment strategies are available anyway [65]. Details about the different rates of re-rupture, complications and infections in the included reviews are shown in Table 3.

These data are also confirmed by the last clinically important paper published on this subject, the RCT performed by Myhrvold and colleagues [66], comparing operative, minimally invasive and non-operative treatments of AATR in 526 patients. They investigated the changes in Total Rupture Score at 12 months, as well as the incidence of tendon re-rupture at 12 months. The different techniques did not significantly differ in Achilles tendon Total Rupture Score, even if the MIS group showed the best improvement at 12 months. Their study also confirmed a significantly higher re-rupture rate for NOT (6%), compared to OR and MIS (0.6% each), and a higher rate of sural nerve injuries in MIS compared to OR (5% vs. 3%). The operative treatment clearly showed fewer re-ruptures, at the cost of a higher complication rate. However, the minimally invasive surgery always resulted in lesser complications and infections, emerging as the best overall treatment strategy in functionally demanding patients.

About the rehabilitation protocols, the early rehabilitation starting within 2 weeks from the operation or cast application, with progressive weight-bearing, shows the best efficacy in terms of return to sports activities. Seow et al. [12] clearly demonstrated the superiority of a rehabilitation initiated within 2 weeks from the trauma, both in operative and nonoperative treated patients, ranging from 0.75 to 0.87 risk ratio for re-ruptures in early vs. late rehabilitation.

## 5. Conclusions

Based on the results of our study, the association of minimally invasive surgery with early rehabilitation protocol results to be the overall best treatment strategy. In elderly patients without high functional demand, but suffering from comorbidities, the non-operative treatment offers a lower risk of complications, paying attention, though, to the higher risk of deep vein thrombosis associated with prolonged immobilization.

## Figures and Tables

**Figure 1 medicina-58-01195-f001:**
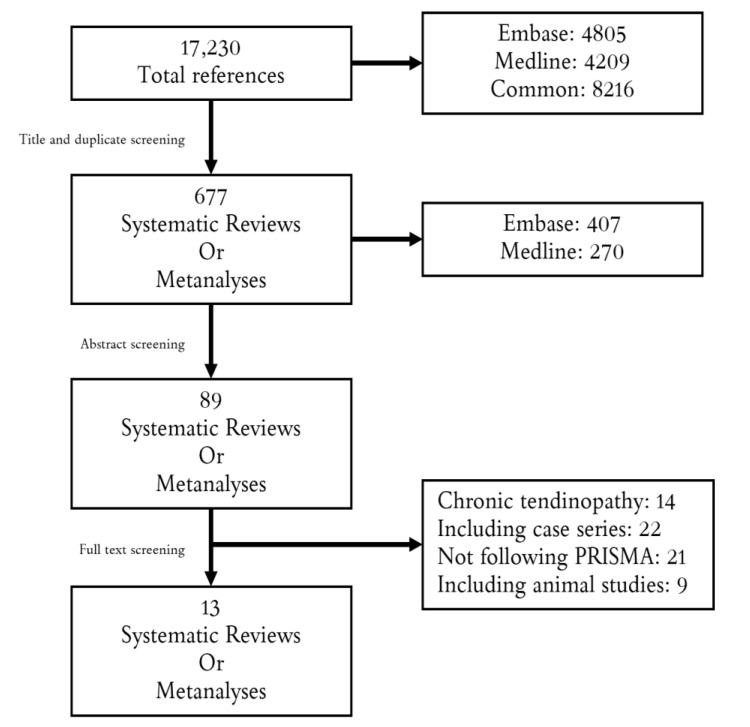
**F****igure****1.** Flowchart of the selection process.

**Figure 2 medicina-58-01195-f002:**
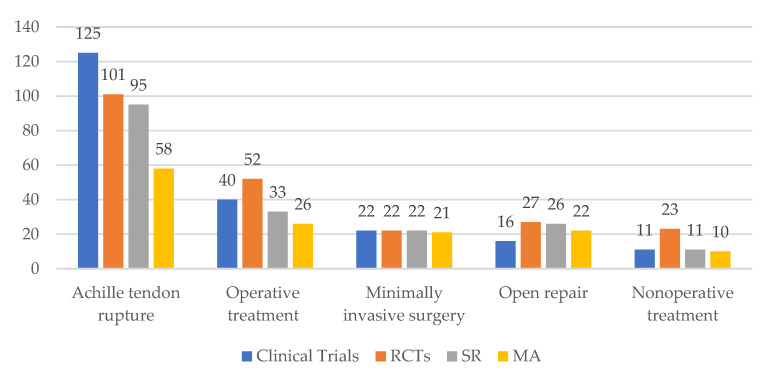
Citations metrics for type of study. Graphic legend: RCT = randomized clinical trials; SR = systematic review; MA = metanalysis.

**Table 1 medicina-58-01195-t001:** Total citations resulted for each keyword searched.

Keyword	Clinical Trials	RCTs	SR	MA	Total Citations
Achille tendon rupture	125	101	95	58	3637
Operative treatment	40	52	33	26	920
Minimally invasive surgery	22	22	22	21	544
Open repair	16	27	26	22	517
Nonoperative treatment	11	23	11	10	153

Table legend: RCT = randomized clinical trials; SR = systematic review; MA = metanalysis.

**Table 2 medicina-58-01195-t002:** Available clinical trials in the literature.

First Author	Comparison	FU	Seow 2021 [12]	Attia 2021 [9]	Shi 2020 [8]	Wu 2018 [11]	Grassi 2018 [10]	Ochen 2018 [14]	Khan 2010 [13]
Nistor 1981 [18]	OR, NOT	30			X	X		X	X
Mortensen 1992 [19]	OR, MIR								X
Saleh 1992 [20]	NOT	12	X						
Cetti 1993 [21]	OR, NOT	12	X		X			X	X
Mortensen 1999 [22]	OR	16	X		X	X			
Majewski 2000 [23]	OR, MIR	30		X	X		X		
Lim 2001 [24]	OR, MIR	6	X	X	X		X		X
Moller 2001 [25]	OR, NOT	24	X		X	X		X	X
Kerkhoffs 2002 [26]	OR	80	X		X				
Kangas 2003 [27]	OR	15	X		X	X			
Costa 2003 [28]	OR	12	X		X				
Costa 2006 [29]	OR	12	X		X				
Twaddle 2007 [30]	OR, NOT	12	X		X	X		X	X
Kangas 2007 [31]	OR	15			X	X			
Gigante 2008 [32]	OR, MIR	12		X	X	X	X		X
Metz 2008 [33]	MIR, NOT	12	x		X	X		X	X
Suchak 2008 [34]	OR	6	X		X	X			
Aktas 2009 [35]	OR, MIR	22	X	X	X	X	X		X
Pajala 2009 [36]	OR	12	X						X
Nilsson-Helander 2010 [37]	OR, NOT	12	X		X	X		X	
Willits 2010 [38]	OR, NOT	24	X		X	X		X	
Keating 2011 [39]	OR, NOT	12	X		X	X		X	
Kołodziej 2012 [40]	OR, MIR	24	X	X	X	X	X		
Karabinas 2013 [41]	OR, MIR	22	X	X	X	X	X		
Olsson 2013 [42]	OR, NOT	12	X		X	X		X	
Schepull 2013 [43]	OR	12	X						
Barfod 2014 [44]	NOT	24	X		X	X			
Groetelaers 2014 [45]	MIR	12	X		X	X			
Young 2014 [46]	NOT	24	X		X	X			
Korkmaz 2015 [47]	NOT	12	X		X				
Porter and Shadbolt 2015 [48]	OR	12	X		X				
Domeij-Arverud 2015 [49]	OR	1.4	X						
Lantto 2016 [50]	OR, NOT	18	X		X	X		X	
De la fuente 2016 [51]	MIR	3	X		X	X			
Heikkinen 2016 [52]	OR	168	X						
Zou 2016 [53]	OR	24	X						
Valkering 2017 [54]	OR	12	X		X	X			
Aisaiding 2018 [55]	OR, MIR	24	X		X	X			
Rozis 2018 [56]	OR, MIR			X	X				
Eliasson 2018 [57]	OR	12	X						
Makulavicius 2019 [58]	OR, MIR	27	X	X	X				
Manent 2019 [59]	OR, NOT	12	X	X					
Kastoft 2019 [60]	NOT	12	X						
Aufwerber 2020 [61]	OR	1.4	X						
Barford 2020 [62]	NOT	12	X						
Costa 2020 [63]	NOT	9	X						

Table legend: The table shows all the available clinical trials in the literature about the treatment of the Achilles tendon rupture (leftmost column) and their inclusion or not in the seven systematic reviews included in our study (right seven columns). In the second column is specified the type of comparison performed in the trial, and in the third column the average follow-up of the patients is included.

**Table 3 medicina-58-01195-t003:** Re-ruptures, complications and infections in different studies.

				Re-Rupture			Complications			Infections		
**Author**	**Included Studies**	**Total Pts**	**Interventions**	**OR**	**MIS**	**NOT**	**OR**	**MIS**	**NOT**	**OR**	**MIS**	**NOT**
Khan 2010 [13]	RCTs	844	OR vs. NOT	5%		12%	18%		0.01%	3.5%		0%
Ochen 2018 [14]	RCTs & OSs	15,862	OR vs. NOT	2.3%		3.9%	4.9%		1.6%	2.8%		0.02%
Grassi 2018 [10]	RCTs	358	OR vs. MIS	0–6%	0–4%							
Wu 2018 [11]	RCTs	2060	OR/MIS/NOT	3%	4%	10%	6%	7%	13%	2%	2%	1%
Shi 2020 [8]	RCTs	2480	OR/MIS/NOT	5%	5%	10%						
Attia 2021 [9]	RCTs	522	OR vs. MIS	2.5%	1.5%		15.5%	10.4%		1.4%	0%	
Seow 2021 [12]	RCTs	-	OR/MIS/NOT	3.3%		12.4%	17.2%	7.7%	3.8%	5.7%	6.2%	0%

Table legends: OR = open repair; MIS = minimally invasive surgery; NOT = nonoperative treatment. Percentages refer to those reported in the relative study by the authors.

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
