# Peer review of "The Acute Achilles Tendon Rupture: An Evidence-Based Approach from the Diagnosis to the Treatment"

_medicina, 2022, doi:10.3390/medicina58091195_

Round 1

Reviewer 1 Report

Even though most orthopaedic surgeons treat Achilles tendon ruptures in their daily practice, still controversies exist concerning treatment or rehabilitation program. This paper shows guidance for treatment of the patients based on literature review. It is well written and concise, which in this type of paper is of high importance in my opinion. I have not found any major flaws concerning study design or paper itself.

Author Response

Dear reviewer, 

We are very grateful to your comments. thank you very much for your time and attention. 

Reviewer 2 Report

It can be accepted after some English revision.

Author Response

Dear Reviewer, 

we made a complete revision of the grammar and modified the text accordingly. 

We thank you very much for your time and comments.

Reviewer 3 Report

The article presents a compendium of information on Achilles tendon rupture, causes, treatment and post-traumatic and treatment effects that is interesting to determine the best technique for rehabilitation. The authors present the most relevant works in the manuscript, however, the abstract presents an excess of words (319). It is suggested to restructure the abstract to 150 words considering the most important without addressing so much the results.

It is suggested to relate the keywords to what is relevant to the research.

In some parts, they write in the first person, it is suggested to change it to the third person.

During the introduction, the citations begin to be placed in correct numerical order (1,2,3, etc.) but the numerical order is lost, it is suggested in this part to order the references being used and place the dates.

- Line 48 starts with parenthesis, but does not close anywhere in the text.

- In line 67 the word PRISMA is not defined.

- In line 87 the word RCTs is not defined.

- In line 109 the values of 0.73 and 0.96 do not specify which units and if not applicable the idea is not understood.

- In line 120 the word Cis is not defined.

Table 1 is shown in a very broad way, it is suggested to reorganize the information, and also to consider the countries of origin of the research that correspond to the authors.

It is recommended that the information on the most relevant keywords be shown visually in a statistical graph to improve the bibliometric study. It would also be good to visualize the number of publications per country on the keywords.

In conclusion, consider information on the statistical study carried out and on the most cited authors on the topic addressed.

Author Response

Dear reviewer, 

thank you very much for your time and comments. 

we modified the abstract according to your indications, and added some keywords related to the topic.

We corrected the numerical order of the references and the minor grammar errors you pointed out. A complete english language revision was performed. 

We added a table and a graph, showing the details of citations obtained for every keywords.

We hope we met your indications. thank you again for your precious comments. 

Round 2

Reviewer 3 Report

Although the content of the paper has been significantly improved by the authors. There are still no statistics with a significant mathematical background with validation and verification that point out the best treatment. The authors should consider adding more explanations with tables and add significant findings to provide more insights into the rehabilitation of the Achilles tendon rupture

The conclusions lack novelty and explanation, new information on the rupture of the Achilles tendon and future work are not mentioned.

Author Response

Dear reviewer,

 thank you again for your time and comments. 

We added a table with the detailed percentages of outcomes included in the studies considered in our review. We also improved the discussion and conclusion following your comments. 

We improved the rehabilitation section, with a detailed reference to the results. 

We hope that our modifications will meet your ideas. 

we look forward to hearing from you, 

kind regards, 

Francesco Amendola